# Minimize Tracking Occlusion in Collaborative Pick-and-Place Tasks: An Analytical Approach for Non-Wrist-Partitioned Manipulators

**DOI:** 10.3390/s22176430

**Published:** 2022-08-26

**Authors:** Hamed Montazer Zohour, Bruno Belzile, Rafael Gomes Braga, David St-Onge

**Affiliations:** 1Department of Mechanical Engineering, École de Technologie Supérieure, Montreal, QC H3C 1K3, Canada; 2Kinova Robotics, Boisbriand, QC J7H 1M7, Canada

**Keywords:** robotics, kinematics, collaborative tasks, occlusion minimization

## Abstract

Several industrial pick-and-place applications, such as collaborative assembly lines, rely on visual tracking of the parts. Recurrent occlusions are caused by the manipulator motion decrease line productivity and can provoke failures. This work provides a complete solution for maintaining the occlusion-free line of sight between a variable-pose camera and the object to be picked by a 6R manipulator that is not wrist-partitioned. We consider potential occlusions by the manipulator as well as the operator working at the assembly station. An actuated camera detects the object goal (part to pick) and keeps track of the operator. The approach consists of using the complete set of solutions obtained from the derivation of the univariate polynomial equation solution to the inverse kinematics (IK). Compared to numerical iterative solving methods, our strategy grants us a set of joint positions (posture) for each root of the equation from which we extract the best (minimizing the risks of occlusion). Our analytical-based method, integrating collision and occlusion avoidance optimizations, can contribute to greatly enhancing the efficiency and safety of collaborative assembly workstations. We validate our approach with simulations as well as with physical deployments on commercial hardware.

## 1. Introduction

Robotic manipulators can be found in a wide range of industrial applications, namely to conduct pick-and-place operations (PPOs). Moreover, the introduction of collaborative robots (cobots) in assembly lines to conduct PPOs usually comes with the need to track the parts with cameras. In these conditions, visual occlusion can reduce productivity and even cause assembly line failures. A system coping with occlusion is essential to reduce potential hazards [1,2]. The goal of this work is to provide a complete automated solution for commercial collaborative manipulators. Indeed, cobots are increasingly used in small- and medium-sized businesses globally every year [3]. This is partially explained by their ease of installation and use, as well as the reduced initial investments required. As explained by Bortolini et al. [4], as part of Industry 4.0, cobots are placed at the core of manufacturing lines. Since they are low-cost, easy to set up, easy to program, and safe to use, they can increase the flexibility of production systems. Moreover, they can expand automation for the purpose of new applications [5].

A symbolic solution to the inverse kinematics problem (IKP) is a powerful tool to achieve versatile control. The vast majority of industrial manipulators with five- or six-revolute joints (commonly referred to as 5R and 6R) are said to be wrist-partitioned (such as the Kuka KR15 and ABB IRB). With these manipulators, we can easily obtain a closed-form solution to their IKP. However, specific conditions must be met so the inverse kinematics of 6R serial manipulators can be decoupled; these conditions are hard to combine with collaborative manipulator characteristics. Conditions on architecture parameters to solve the orientation and positioning problem separately often lead to a wrist joint analog to a spherical configuration [6]. This condensed configuration of the wrist is too complex to fully enclose, e.g., it prevents any finger of a user to be trapped or pinched. This presents a major limitation for collaborative operations. Some robots, such as the Kinova Gen3 series represented in Figure 1 by the Gen3 lite model, are designed to optimize the safety and reachable workspace but are more complex non-wrist-partitioned manipulator architectures.

The complexities of the kinematics of these manipulators steer most of the research energy over collision-free trajectory planning [7,8] and compliant grasping techniques [9]. The numerical solvers used in these works do not provide enough flexibility and cannot guarantee that a solution will be found. In this work, we proposed a complete solution to minimize tracking occlusion for collaborative pick-and-place tasks.

On the perception side of the problem, several works cope with adaptive sensing for object tracking with occlusion. Many consider *occlusions inevitable*, and rather suggest approaches based on multimodal sensing, for instance with sensitive finger tips [1]. Learning strategies may also help reconstruct known objects from partial images [2]. With an occlusion-free strategy, these works will serve to enhance the robustness to occlusion from the environment only. As this work calls for several bodies of knowledge, a review of the related work is divided into subcategories in the following section.

The flow chart detailing the procedure is shown in Figure 2. In block 1, we detect the object to grasp as detailed in Section 6. If it fails, we move the camera (block 2) hook as a robotic manipulator end-effector to another pose (Section 6). If the camera is able to detect the target, its pose is used to compute the IK polynomial roots (block 3). We leveraged a methodology first introduced by Gosselin and Liu [10] to obtain a univariate polynomial equation for the IKP (Section 3 and Section 4), giving us all possible postures (joint space) of the robot for the same end-effector pose (Cartesian space). Section 8.1 presents a validation of the IKP solution with examples and compares it with the solutions obtained with a numerical IKP solver. In block 4, we select the best solution to avoid the occlusion between the actuated camera and all known objects in the workspace (Section 5). Among these obstacles, the operator is tracked by the camera (block 5, Section 6). Finally, a path-planner is used to compute the trajectory between the original and final (optimal) posture, taking into account the occlusion and obstacles (block 6, Section 7). If the planning fails, we change the camera point-of-view (block 2), and start the previous steps again. The proposed methodology was validated in a simulation and with experiments (Section 9). The Python script used to solve the IKP, compute all real solutions (postures), and select the optimal posture is available online [11].

## 2. Related Work

### 2.1. Full-Stack Solutions and Industrial Approaches

Transitioning from the standard isolated industrial robots setup to the shared space and task between workers and robots brings several safety risks and performance concerns. It is often considered difficult to automate an assembly process crowded with workers, product parts, and tools. Therefore, Cherubini et al. [12] observed that collaborative robotic systems have generated interest in heavy and laborious tasks while allowing humans to work on more value-added tasks. Hanna et al. [13] studied an industrial application for which autonomous robotic systems were alongside manual assembly lines. They highlight several challenges and requirements related to worker safety, intuitive interactions, adaptations to variable processes, and the need for highly flexible communication and control. Similar results were discussed by Marvel et al. [14]. Using the manual assembly station from that study as a starting point, Hanna et al. [15] then focused on the safety aspects of a cobot station. While a trained worker can safely use a complex and ultimately dangerous robotic system, the transition of the industry requires additional safety measures. In this context, a robot integrator must think outside the current standards and guidelines. Hanna et al. [15] suggested a new collaboration mode: deliberative planning and acting. To support the transition, Ogorodnikova [16] introduced danger/safety indices that indicate the level of risk during an interaction with a robotic system. The indices are based on the characteristics of the robot and the operator’s physical and cognitive capacities. He stressed that the system must be intuitive and easy to use for the worker in order to be safe. With this in mind, we designed an interaction modality centered on worker safety, which does not increase the actions (commands) of the worker.

The vast majority of research and industrial use case studies on the transition from manual to cobot-equipped assembly stations present the need to reduce the physical risks to the worker. These studies, such as the one by Salunkhe et al. [17], focused on decreasing ergonomic issues, maintaining or decreasing cycle times at the station, and maintaining or increasing product quality. However, they do not cover how the workers interact with the system, rather, they split the tasks and maintain (close) distinct workspaces. On the hardware side, cobots can be designed specifically for safe collaboration, such as UR10 [18] and KUKA iiwa [19]: they detect collisions with any part of their structures, carry smaller loads, and have shorter reaches. The last two attributes may enhance safety, but they limit their application. Gopinath et al. [20] argue that close collaboration with large industrial robots can be safe and they show two experimental workstations. The key is to better understand the task and the operator and, thus, how to make stations safe. Smart control strategies tailored to a good understanding of the application is what Shadrin et al. [21] leveraged to increase safety by modeling the objects and environment.

Researches have demonstrated the need for smart adaptive solutions to human–robot collaboration(s) (HRC) in assembly lines. Our solution provides a flexible and optimal way to avoid any collision and ensure a safe collaboration.

### 2.2. Serial Manipulators

As respectively shown by Pimrose [22] and Lee et al. [23], a general 6R robotic manipulator has a maximum number of 16 different solutions to its IKP for a given end-effector pose. A polynomial degree of 16 is the lowest possible that can be obtained for a univariate polynomial equation describing the kinematics of the robot. Polynomial solutions for different manipulators can be found in the literature [10,24] with similar methodologies as the one described in this work. Considering that 16th-degree polynomial equations are prone to numerical ill-conditioning as well as the possibility of *polynomial degeneration* with roots yielding angles of π, Angeles and Zanganeh proposed a semi-graphical solution to the inverse kinematics of a general 6R serial manipulator [25]. However, these techniques do not apply to non-wrist-partitioned manipulators. Numerical methods have also been applied by several researchers [26,27,28], but these are commonly known to be prone to instability near singular postures. Moreover, they only give one possible solution, which may not be optimal. Several algorithms, including the ones proposed by Mavroidis et al. [29], Husty et al. [30], and Qiao et al. [31], can be found in the literature to find the 16th-degree univariate polynomial equation for a 6R robotic manipulator, the latter notably using double quaternions.

### 2.3. Optimal Solution to the IKP

Symbolic solutions to the IKP, such as the one presented above, mostly result in several viable configurations for a given end-effector pose. Thus, a strategy is required to select the best-fitted solution; a single set of joint angles. A wide range of procedures can be used to select the optimal solution following the task (such as manipulating fragile objects) and the application context (such as low energy requirements).

Among the 16 solutions to the IKP, a wide range of methodologies has been proposed to select the best posture. As these solutions are theoretical, one must first discard the one that cannot be implemented: non-real roots exceeding joint limits or resulting in a self-colliding posture. From there, simple algorithms, such as the minimization of the number of joint rotations, can easily be implemented. Task-dependent optimization can also be used for certain applications and performance indices based on the kinematics (e.g., kinetostatic conditioning index) and the stiffness (e.g., deformation evaluation index) of the robot [32]. Guo et al. [33] used a method based on the Jacobian matrix to solve the robot posture optimization model with the aim of increasing the stiffness of the robot in machining applications. Zargarbashi et al. [34] reported posture-dependent indices based on kinetostatics to optimize the posture of a redundant robot for the given task. Our approach is task-related: preventing camera occlusions for pick-and-place tasks.

### 2.4. Trajectory Planning

From the set of joint angles for the manipulator goal, we need to derive the optimal path. These motions are typically synthesized to achieve functional goals, such as minimizing time, maximizing efficiency, and providing sufficient clearance around obstacles. Lozano-Perez et al. [35] were among the firsts to use the concept of task planning; since then, a large range of algorithms have been proposed. The initial focus of motion planning research is concentrated on finding a complete planning algorithm, where an algorithm is said to be complete if it terminates in finite time, returning a valid solution if one exists, and failure otherwise. Early work focused on finding trajectories that satisfy constraints imposed by the environment of the application, but they were not necessarily optimal. Yang et al. [36] provided a selection of optimal motion planning algorithms studied in terms of three main components: the decision variables, constraints, and objectives. The two most influential families of path planners are the sampling-based algorithms [37,38,39] and the optimization-based ones [40,41,42]. While the former is often more efficient for collision avoidance, the latter grants more flexibility on the optimization criteria.

To plan the trajectory for robots with high degrees of freedom (DoFs), such as industrial robots (usually six or seven DoFs) and mobile manipulators (usually more than seven DoFs), one main contribution to the motion planning field involves the development of sampling-based algorithms [37]. The sampling-based planning algorithm is one of the most powerful tools for collision avoidance. Moreover, planners, such as probabilistic roadmap (PRM) and rapidly-exploring random tree (RRT) algorithms, along with their descendants, are now used in a multitude of robotic applications [37,38]. Both algorithms are typically deployed as part of a two-phase process: first, find a feasible path, and then optimize it to remove redundant or jerky motion. Study [39] proposed a goal-oriented (GO) sampling method for the motion planning of a manipulator.

In that second family, Ratliff et al. [40] proposed the covariant Hamiltonian optimization for motion planning (CHOMP): a novel method for generating and optimizing trajectories for robotic systems. Unlike many previous path optimization techniques, the requirement that the input path be collision-free was dropped. Kalakrishnan et al. [41] presented the stochastic trajectory optimization for motion planning (STOMP) using a series of noisy trajectories that can deal with general constraints. Otherwise, Park et al. [42] developed a novel algorithm to compute real-time optimization-based collision-free trajectories in dynamic environments without the requirement for prior knowledge about the obstacles or their motions. These algorithms and several others were integrated into the Open Motion Planning Library (OMPL) [43]. Our solution leverages these powerful contributions.

## 3. Dual-Arm Configuration

The assembly workstation we designed consists of an operator and a robotic arm helping the operator with picking specific parts. To support the collaboration, we added a second robotic arm controlling the camera point-of-view. For staging the experiments, we designed small cubes with fiducial markers as the parts (target objects). An overview of the experimental setup, discussed in Section 9, is shown in Figure 3.

## 4. 6R Cobot Kinematics

### 4.1. Manipulator

As an example of a non-wrist-partitioned cobot, we tailored our derivation to Kinova Gen3 lite, a serial manipulator with six revolute joints each having limited rotation and a two-finger gripper as the end-effector (EE). The Denavit–Hartenberg (DH) parameters of this robot are given in Table 1 (numerical values given in Section 8.1), where the non-zero parameters are identified. With the parameters in this table, it is clear that this robot is not wrist-partitioned since $b_{5}\neq 0$. Thus, well-known methodologies to find the decoupled solution of the IKP cannot be used.

As shown in Figure 1, a DH reference frame is attached to each link. It should be noted that these frames are not necessarily located at the joints. The rotation matrices $Q_{i}$ and the position vectors $a_{i}$ related to the successive reference frames defined on each of the links of the robot [29] can be written as
(1a)$$Q_{i}=\left[\begin{array}{ccc} \cos\theta_{i} & -\cos\alpha_{i}\sin\theta_{i} & \sin\alpha_{i}\sin\theta_{i}\\ \sin\theta_{i} & \cos\alpha_{i}\cos\theta_{i} & -\sin\alpha_{i}\cos\theta_{i}\\ 0 & \sin\alpha_{i} & \cos\alpha_{i} \end{array}\right]$$
(1b)$$a_{i}={\left[\begin{array}{ccc} a_{i}\cos\theta_{i} & a_{i}\sin\theta_{i} & b_{i} \end{array}\right]}^{T}$$
where the rotation matrix $Q_{i}$ rotates frame *i* into the orientation of frame (i+1) and the vector $a_{i}$ connects the origin of frame *i* to the origin of frame (i+1). The joint variables are noted $\theta_{i}$ while $a_{i}$, $b_{i}$ and $\alpha_{i}$ are the DH parameters representing the geometry of the Kinova Gen3 lite. The end-effector is located at the origin of frame 7, which is defined by the three-dimensional vector p. The orientation of the end-effector is given by the rotation matrix from frame 1 to frame 7, noted as Q.

### 4.2. IKP Analytical Solution

The forward kinematic problems (FKPs), i.e., the Cartesian position p and orientation matrix of the tool Q, are straightforward and can be written as
(2)$$p=\sum_{i=0}^{5}\left(\left(\prod_{j=0}^{i}Q_{j}\right)a_{i+1}\right),\;Q=\prod_{i=1}^{6}Q_{i}$$
where $Q_{0}$ is the (3×3) identity matrix. The first step toward solving the IKP of the Gen3 lite is to reduce the number of unknowns, currently six for the six joint positions $\left\{\theta_{i}\right\}$, to one, reducing the problem to a univariate polynomial equation that can be solved. By finding expressions for $\sin\theta_{i}$ and $\cos\theta_{i}$ and substituting them in $\sin^{2}\theta_{i}+\cos^{2}\theta_{i}=1$, we can readily reduce the number of unknowns. First, we need to compute r, connecting the origin of frame 1 to the origin of frame 6, which can be written similarly to the left-hand part of Equation (Equation 2) as
(3)$$r=\sum_{i=0}^{4}\left(\left(\prod_{j=0}^{i}Q_{j}\right)a_{i+1}\right)$$

By pre-multiplying Equation (Equation 3) by $Q_{1}^{T}$ and isolating all expressions independent of $\theta_{1}$ on the right-hand side, we have a set of three scalar equations. Among them, two stand out as only being functions of $\theta_{1}$, $\theta_{2}$ and $\theta_{\left(3-2\right)}$: (4)$$\begin{array}{cc} r_{1}c_{1}+r_{2}s_{1} & =a_{2}c_{2}+b_{5}c_{\left(3-2\right)}s_{4}+b_{4}s_{\left(3-2\right)} \end{array}$$(5)$$\begin{array}{cc} r_{3}-b_{1} & =a_{2}s_{2}-b_{5}s_{\left(3-2\right)}s_{4}+b_{4}c_{\left(3-2\right)} \end{array}$$
where $r_{i}$ is the *i*th component of r, $s_{i}$, $c_{i}$, $c_{\left(i-j\right)}$ and $s_{\left(i-j\right)}$ stand, respectively, for $\sin\theta_{i}$, $\cos\theta_{i}$, $\cos(\theta_{i}-\theta_{j})$ and $\sin(\theta_{i}-\theta_{j})$. The last scalar equation remaining after pre-multiplying Equation (Equation 3) by $Q_{1}^{T}$ and will be needed later in the derivation:(6)$$r_{1}s_{1}-r_{2}c_{1}=b_{2}-b_{3}+b_{5}c_{4}.$$
which can be rewritten to obtain an expression of $c_{4}$:(7)$$c_{4}=\left(r_{1}s_{1}-r_{2}c_{1}+b_{3}-b_{2}\right)/b_{5}$$

We are now able to solve Equations (Equation 4) and (Equation 5) for $s_{2}$ and $c_{2}$. Substituting the results in $s_{2}^{2}+c_{2}^{2}=1$, we obtain
(8)$$B_{1}s_{\left(3-2\right)}+B_{2}c_{\left(3-2\right)}+B_{3}=0$$
where $B_{1}$, $B_{2}$ and $B_{3}$ are functions of the DH parameters and $c_{1}$, $s_{1}$ and $s_{4}$.

Having a first equation expressed as a function of $s_{\left(3-2\right)}$ and $c_{\left(3-2\right)}$, a second one is needed to compute $s_{\left(3-2\right)}^{2}+c_{\left(3-2\right)}^{2}=1$. Matrices Q being orthogonal matrices, the right-hand part of Equation (Equation 2) can be recast into
(9)$$Q_{4}Q_{5}Q_{6}=Q_{3}^{T}Q_{2}^{T}Q_{1}^{T}Q$$

This equation gives us a system of nine scalar equations. However, only five are relevant, with the ones defining the first two components of the last row and the three components of the last column of the resulting matrices. On the one hand, the former can be used to obtain expressions of $c_{6}$ and $s_{6}$:
(10a)$$\begin{array}{cc} c_{6} & =\left(q_{11}c_{1}s_{\left(3-2\right)}+q_{21}s_{1}s_{\left(3-2\right)}+q_{31}c_{\left(3-2\right)}\right)/s_{5} \end{array}$$
(10b)$$\begin{array}{cc} s_{6} & =\left(q_{12}c_{1}s_{\left(3-2\right)}+q_{22}s_{1}s_{\left(3-2\right)}+q_{32}c_{\left(3-2\right)}\right)/-s_{5} \end{array}$$

These two equations will be useful later in the paper. On the other hand, the components of the last column are not a function of $\theta_{6}$, because the latter corresponds to a rotation of the last joint about the *z*-axis of the end-effector. Therefore, the last column, defining a unit vector parallel to this axis, must be independent of $\theta_{6}$. With this column, we obtain the following scalar equations, which are cast in an array form with dialytic elimination:
(11a)$$Mk_{5}=0$$
where 0 is a three-dimensional zero vector and
(11b)$$M=\left[\begin{array}{ccc} 0 & -c_{4} & m_{13}\\ 0 & -s_{4} & m_{23}\\ 1 & 0 & m_{33} \end{array}\right],\;k_{5}=\left[\begin{array}{c} c_{5}\\ s_{5}\\ 1 \end{array}\right]$$
with, after some simplifications,
(11c)$$\begin{array}{cc} m_{13} & =\left(q_{13}c_{1}+q_{23}s_{1}\right)c_{\left(3-2\right)}-q_{33}s_{\left(3-2\right)} \end{array}$$
(11d)$$\begin{array}{cc} m_{23} & =(-q_{13}s_{1}+q_{23}c_{1}) \end{array}$$
(11e)$$\begin{array}{cc} m_{3} & =\left(q_{13}c_{1}+q_{23}s_{1}\right)s_{\left(3-2\right)}+q_{33}c_{\left(3-2\right)} \end{array}$$

In the above expressions, $q_{ij}$ is the (i, j)th component of the end-effector orientation matrix Q. It can be seen that M, a homogeneous matrix, in Equation ([Disp-formula FD11a-sensors-22-06430]), is singular, as vector $k_{5}$ cannot vanish. Therefore, we have
(12)$$\det\left(M\right)=A_{1}s_{\left(3-2\right)}+A_{2}c_{\left(3-2\right)}+A_{3}=0$$
where $A_{1}$, $A_{2}$ and $A_{3}$ are functions of the EE pose, $\theta_{1}$ and $\theta_{4}$ only. Equations (Equation 8) and (Equation 12) can now be solved for $s_{\left(3-2\right)}$ and $c_{\left(3-2\right)}$, and substituted in $s_{\left(3-2\right)}^{2}+c_{\left(3-2\right)}^{2}=1$, yielding
(13a)$$\begin{array}{cc} c_{\left(3-2\right)} & =\left(A_{3}B_{1}-B_{3}A_{1}\right)/\left(B_{2}A_{1}-A_{2}B_{1}\right) \end{array}$$
(13b)$$\begin{array}{cc} s_{\left(3-2\right)} & =\left(A_{3}B_{2}-B_{3}A_{2}\right)/\left(B_{2}A_{1}-A_{2}B_{1}\right) \end{array}$$
and, finally,
(13c)$$\begin{array}{c} {\left(A_{2}B_{3}-A_{3}B_{2}\right)}^{2}+{\left(A_{3}B_{1}-A_{1}B_{3}\right)}^{2}\\ -{\left(A_{1}B_{2}-A_{2}B_{1}\right)}^{2}=0 \end{array}$$

Having eliminated all expressions of $\theta_{2}$ and $\theta_{3}$ with the procedure above, Equation ([Disp-formula FD13c-sensors-22-06430]) is only a function of $\theta_{1}$ and $\theta_{4}$, bringing us closer to our objective of finding a univariate polynomial equation. Equation ([Disp-formula FD13c-sensors-22-06430]) can be factorized as a function of powers of $c_{4}$ and $s_{4}$, giving us
(14)$$\begin{array}{c} F_{1}c_{4}^{6}+F_{2}c_{4}^{5}+F_{3}c_{4}^{4}+F_{4}c_{4}^{3}s_{4}+F_{5}c_{4}^{3}+F_{6}c_{4}^{2}s_{4}\\ +F_{7}c_{4}^{2}+F_{8}c_{4}s_{4}+F_{9}c_{4}+F_{10}s_{4}+F_{11}=0 \end{array}$$
where the coefficients $F_{i},\;i=1,\cdots ,11$ are solely dependent of $\theta_{1}$. With Equation (Equation 7), Equation (Equation 14) becomes
(15a)$$Vs_{4}+W=0$$
with
(15b)$$\begin{array}{c} \begin{array}{cc} V= & v_{1}c_{1}^{3}+v_{2}c_{1}^{2}s_{1}+v_{3}c_{1}^{2}+v_{4}c_{1}s_{1}+v_{5}c_{1}\\ \; & +v_{6}s_{1}+v_{7} \end{array} \end{array}$$
(15c)$$\begin{array}{c} \begin{array}{cc} W= & w_{1}c_{1}^{4}+w_{2}c_{1}^{3}s_{1}+w_{3}c_{1}^{3}+w_{4}c_{1}^{2}s_{1}+w_{5}c_{1}^{2}\\ \; & +w_{6}c_{1}s_{1}+w_{7}c_{1}+w_{8}s_{1}+w_{9} \end{array} \end{array}$$
where $v_{i}$ and $w_{i}$ are only functions of the DH parameters and the orientation Q and position p of the tool. The above equation can be solved for $s_{4}$, then substituted, with Equation (Equation 7) in $s_{4}^{2}+c_{4}^{2}=1$. The resulting univariate equation is
(16)$$b_{5}^{2}W^{2}+\left[{\left(r_{1}s_{1}-r_{2}c_{1}+b_{3}-b_{2}\right)}^{2}-b_{5}^{2}\right]V^{2}=0$$

Equation (Equation 16) is one of degree 8 in terms of $c_{1}$ and of degree 1 in terms of $s_{1}$. Then, using the Weierstrass substitution, Equation (Equation 16) is finally transformed into a polynomial in $T_{1}=\tan\left(\theta_{1}/2\right)$:(17)$$\sum_{i=0}^{16}E_{i}T_{1}^{i}=0$$
where $\left\{E_{i}\right\}$ are functions of the DH parameters and the pose of the end-effector of the manipulator at hand. The roots of this univariate polynomial can then be computed to obtain $T_{1}$, leading to the values of $\theta_{1}$. Some of these solutions may be complex numbers and some can be duplicates. For control, only the real roots can be considered. Using a subset of the equations presented above, it is possible to compute all other joint angles for each real solution. For all remaining joint angles, a single trigonometric function is needed, i.e., $\theta_{i}=\arctan2\left(s_{i},c_{i}\right)$. The equation numbers for expressions of $s_{i}$ ($\sin\theta_{i}$) and $c_{i}$ ($\cos\theta_{i}$) are given in Table 2. The back substitution procedure must be conducted following the order from left to right and top to bottom presented in this table, starting with $c_{4}$. Finally, $\theta_{3}$ is easily computed from $\left(\theta_{3}-\theta_{2}\right)$ and $\theta_{2}$.

### 4.3. Special Cases

Similar to the majority of similar algorithms, some special cases must be considered. The special cases considered here are similar to those pointed out by Gosselin and Liu [10] for another manipulator; their methodology can also be applied to this manipulator.

First, it is possible that coefficient *V* in Equation ([Disp-formula FD15a-sensors-22-06430]) becomes equal to zero. Since, according to the procedure detailed in the previous section, both $s_{4}$ and $c_{4}$ are required, the value of $\theta_{4}$ cannot be computed with *atan2*. Instead, *arccos* must be used, and two values of $\theta_{4}$ for a single $\theta_{1}$ will be obtained. Of course, since the total number of solutions cannot exceed 16, some will be repeated. Another possible special case arises when $\left(B_{2}A_{1}-A_{2}B_{1}\right)$ is equal to zero. Thereby, Equations ([Disp-formula FD13a-sensors-22-06430]) and ([Disp-formula FD13b-sensors-22-06430]) cannot be computed. Instead, Equations (Equation 8) and (Equation 12) are solved for $\theta_{\left(3-2\right)}$ with the Weierstrass substitution previously mentioned, leading to two solutions for $\theta_{\left(3-2\right)}$ for a single $\theta_{1}$. As always, no more than 16 unique sets of joint angles can be obtained, which means there will be some repeated solutions again.

## 5. Optimal Occlusion Avoidance Posture

After obtaining the solutions to the IKP for a particular end-effector pose for a pick-and-place task, the next step consists of selecting the optimal solution, as highlighted in the flowchart illustrated above (Figure 2). This is done in two sub-steps, namely reducing the number of solutions to the one respecting an occlusion avoidance threshold, then choosing the one with the shortest path, as we detail in the following section.

A common setup for pick-and-place tasks is to rely on a top-view camera, positioned above the table workspace. The optimization criterion is to maximize the field of view up to a certain threshold. This can be extended to several pick-and-place operations. Thus, the objective is to avoid the manipulator interfering with the camera’s line of sight to the objects on the table. To this aim, simple line geometry is used and the shortest distance between all links and the line of sight with all objects is computed, as depicted in Figure 4. To avoid occlusion, the latter must be kept above an arbitrary threshold $d_{th}$, i.e., there is no need to maximize it. We determined a unique threshold from the length of the largest link radius plus a buffer distance.

First, the position of a point along the straight line P from the camera, located at $O_{p}$, to a small object, located at $O_{z}$, is defined as
(18)$$s_{i}=O_{p}+\Delta_{p,i}\left(O_{z}-O_{p}\right)$$
where $\Delta_{p,i}$ is a factor defining where along the line this point is located. Moreover, the Cartesian coordinates of points $S_{i}$, $O_{p}$ and $O_{z}$ are, respectively, arrayed in vectors $S_{i}$, $O_{p}$ and $O_{z}$. Similarly, the position of a point $P_{i}$ along the line $L_{i}$ can be defined for any given link of the manipulator, i.e.,
(19)$$p_{i}=O_{i}+\Delta_{i}\left(O_{i+1}-O_{i}\right),\;i=2,\cdots ,8$$
where $O_{i}$ and $\Delta_{i}$ are, respectively, the Cartesian coordinates of the intersections between the links and a factor defining where along the latter this point is located. If these two points are the closest pair along their respective lines, a unit vector, orthogonal to $L_{i}$ and P, thus parallel to $D_{i}$, can be defined as
(20)$$v_{i}=\frac{\left(O_{z}-O_{p}\right)\times \left(O_{i+1}-O_{i}\right)}{\left|\right|\left(O_{z}-O_{p}\right)\times \left(O_{i+1}-O_{i}\right)\left|\right|}$$

Vectors $\left\{O_{i}\right\}$ should not be confused with the locations of the DH frames, i.e., $\left\{p_{i}\right\}$. Moreover, the first link, which is rigidly attached to the base, is not considered, since it does not translate. With these three vectors ($s_{i}$, $p_{i}$, $v_{i}$), a close loop equation is formulated:(21)$$s_{i}=p_{i}+\Delta_{d,i}v_{i}$$
where $\Delta_{d,i}$ is the shortest distance between $L_{i}$ and P. Thus, a set of three linear equations with three unknowns, $\Delta_{i}$, $\Delta_{p,i}$ and $\Delta_{d,i}$, is obtained and can easily be solved.

The value of these three unknowns obtained and the risk of occlusion for an object on the table can now be computed. Indeed, the shortest distance between the robot and $\bar{O_{p}O_{z}}$, namely $\min(\Delta_{d,1},\cdots ,\Delta_{d,6})$, for a prescribed end-effector position and orientation must be larger than a certain threshold. Of course, if point $P_{i}$ for a robot posture and a given link is not located within the limits of the latter, the corresponding $\Delta_{d,i}$ should be disregarded. It is the case, for instance, when $O_{p}$, $O_{i}$, and $O_{i+1}$ are aligned. Instead, the closest distance between a line ($\bar{O_{p}O_{z}}$) and a point (the corresponding link end) should be computed. This is done with the following equations:
(22a)$$\begin{array}{cc} \Delta_{d,i}= & \frac{\left|\right|\left(O_{p}-O_{i}\right)\times \left(O_{z}-O_{p}\right)\left|\right|}{\left|\right|O_{z}-O_{p}\left|\right|},\;\mathrm{if}\Delta_{i}<0 \end{array}$$
(22b)$$\begin{array}{cc} \Delta_{d,i}= & \frac{\left|\right|\left(O_{p}-O_{i+1}\right)\times \left(O_{z}-O_{p}\right)\left|\right|}{\left|\right|O_{z}-O_{p}\left|\right|},\;\mathrm{if}\Delta_{i}>1 \end{array}$$

The procedure above is valid if the object is relatively small, i.e., with external dimensions smaller than the threshold chosen. If it is not the case, the proposed technique can still be adapted. Indeed, the line of sight between each object in the workspace and the top-view camera is instead modeled as an irregular pyramid. Therefore, instead of having only one line $\bar{O_{z}O_{p}}$ for each object, the periphery of the latter, as seen by the camera, is discretized (with a step size smaller than $2d_{th}$), as depicted in Figure 5. Therefore, the distance between each link of the robot and each line defining each pyramid must be computed for each feasible final posture. In this way, the equations detailed above can be used without any modification.

## 6. Target and Operator Tracking

The top-view camera configuration mentioned in the previous section is a common choice for assembly tasks without an operator. However, they are prone to occlusion by the operator’s head and torso whenever he/she bends over the table. Ultimately, no fixed camera position can provide a guarantee of keeping a line of sight on the target. To easily control the pose of the camera in the 3D space, we rigidly attached it to another identical robot. Thus, we have two Kinova Gen3 lite manipulators. As can be seen in Figure 3 (top), an Intel Realsense D455 camera is mounted on the second manipulator’s end-effector, visible on the left-hand side of the figure. Any other robotic arms equipped with a wrist camera can be used for this purpose (for instance the Gen3 and the RobotiQ wrist sensors). We then compute the geometrical transformation from the camera to the right end-effector frame (worker robot) using its forward kinematics (described in Section 4). Frames are illustrated in Figure 6.

The pose of the object (cube) to be picked is obtained using an Apriltag marker attached to it. The Realsense camera detects the tag in the camera frame and we then project it in the worker arm reference frame (right-hand side of Figure 6). As mentioned in the previous section, we add a *vision* cone from the tag to the camera in the virtual workspace as an obstacle to avoid occlusion.

As for detecting the operator—we leverage the Nuitrack software [44] fed with the same camera (D455, RGB, and depth). Nuitrack’s AI skeleton tracking feature provides full body skeleton tracking based on RGB-D data, as illustrated in Figure 7. Using Nuitrack’s SDK, we extract the Cartesian coordinates of the operator’s shoulders and elbows and broadcast them as ROS topics. Since the arms are considered obstacles, another node catches the topic and generates cylinders in the virtual workspace that can be visualized in Rviz, as shown in Figure 3 (bottom).

The camera stays still most of the time unless it cannot detect the target or the path planner fails. In this case, we change the camera pose, which leads to generating another occlusion cone to avoid and, thus, a different path planning. Currently, several manually tuned camera poses have been recorded and selected randomly.

## 7. Shortest Occlusion-Free Path

After the initial stage, described in Section 5, where the set of feasible solutions to the IKP is reduced to only the postures respecting the occlusion avoidance criterion, the next step is to plan the trajectory, and, most importantly, to select the shortest path. This can be done with any trajectory planner, such as the ones available in the OMPL, integrated into ROS MoveIt!. Similar to the section above, the line of sight between each object in the workspace and the variable-pose camera is modeled as an irregular pyramid as shown in Figure 8. The apex of the latter is found at the location of the camera. These pyramids are included in the environment as virtual obstacles to be taken into account by the trajectory planner, which is fed with the postures where the shortest distance with the virtual obstacles is above the defined occlusion threshold to select the one with the shortest path.

## 8. Examples and Numerical Validation

### 8.1. IKP Solutions

This section presents two examples to illustrate the IKP presented above. A Python script was written to process all equations and is publicly available online [11]. It should be noted that while some solutions may be theoretically possible, they are not feasible in practice because of the mechanical limits of the joints. The numerical values of the DH parameters and the joint limitations are given in Table 3. Finally, the roll–pitch–yaw angles are used to give the orientation of the end-effector. Incidentally, the orientation matrix Q is defined as
(23)$$Q\equiv \left[\begin{array}{ccc} c_{\psi }c_{\theta } & -s_{\psi }c_{\phi }+c_{\psi }s_{\theta }s_{\phi } & s_{\psi }s_{\phi }+c_{\psi }s_{\theta }c_{\phi }\\ s_{\psi }c_{\theta } & c_{\psi }c_{\phi }+s_{\psi }s_{\theta }s_{\phi } & -c_{\psi }s_{\phi }+s_{\psi }s_{\theta }c_{\phi }\\ -s_{\theta } & c_{\theta }s_{\phi } & c_{\theta }c_{\phi } \end{array}\right]$$
where ϕ, θ and ψ are the roll, pitch, and yaw angles, respectively, and $c_{\gamma }\equiv \cos\gamma$, $s_{\gamma }\equiv \sin\gamma$, for γ={ϕ,ψ,θ}.

### 8.2. Validation of the IKP Solution Selection Criterion

For this first example, the position and orientation of the end-effector are detailed in Table 4. The obtained solutions are shown in Figure 9a. It should be noted that 10 solutions were initially found by solving the IKP; however, only 6 were within the joint limitations, detailed in Table 5.

We also included in Table 5 the numerical solutions obtained with the ROS MoveIt! IK package and with the robot numerical IK embedded controller, both being among the solutions obtained with the procedure detailed in Section 4.

In our second example, we simulate a pick-and-place task. To grasp the object, the position and orientation of the end-effector were first determined, as detailed in Table 4. Then our IKP script was used, leading to a set of eight solutions illustrated in Figure 9b. The four solutions respecting the joint limitations are detailed in Table 6, as well as the numerical solution obtained with ROS MoveIt! IK and the robot numerical IK embedded controller. Excerpts of two solutions are depicted in Figure 9c,d from the ROS-Gazebo simulation. They will be used in the next section to illustrate the selection of the optimal posture.

With the postures presented in Table 6, solution no. 8, depicted in Figure 9b, is one of the potential final postures identified by the algorithm as respecting the criterion, i.e., the shortest distance is above the threshold defined with two small objects that must remain visible to the camera above the workspace (with $O_{p}={\left[0.32\;0\;1.3\right]}^{T}$ m, $O_{z,1}={\left[0.25\;0.15\;-0.002\right]}^{T}$ m, and $O_{z,2}={\left[0.25\;-0.15\;-0.002\right]}^{T}$ m). In this case, considering a threshold of $d_{th}=6$ cm, which is larger than the objects’ diameter of 3 cm, only one line for each is considered for the line of sight. The smallest distance between the robot and any of the two is, in this example, 0.0972 m. Moreover, this test was validated experimentally, as shown in Figure 10. The photos are taken from the camera located at $O_{p}$, showing clearly that solution no. 8 is significantly better than solution no. 6 with respect to the occlusion risks for objects located at $O_{z,1}$ (top) and $O_{z,2}$ (bottom). Solution no. 7 is the only other feasible posture with the shortest distance above the threshold of $d_{th}=0.06$ m.

### 8.3. Validation of the Path Planner

As a first validation step, we recall example no. 2 from Section 8.1, for which the optimal solution was already detailed among the set of possible solutions. The line of sight obstacle cones were modeled in MoveIt!. For each feasible (and occlusion-free) final posture, the trajectory planner is run to find the shortest path avoiding these virtual obstacles. Finally, among the solutions found, the shortest path is the optimal occlusion-free solution. Here, among the remaining feasible occlusion-free solutions, no. 8 has the shortest path with 2.22 m (2.53 m for solution no. 7) and, therefore, is our optimal solution.

## 9. Experiments

To test and compare the performance of our proposal, six other test cases, illustrated in Figure 11 were executed. For each test case, the two cubes were positioned in different locations within the workspace and the operator position changed slightly. A sample of the test configuration is illustrated in Figure 12. As it was done in the previous scenario, virtual obstacles representing the line of sight between the camera and the objects are included, with the addition of the operator’s tracked arms. Again, the shortest path between the solutions above the occlusion threshold is chosen.

While several solutions to the IKP may result in an occlusion-free final posture (respecting the threshold), the path planner must adapt the trajectory. The bottom configuration sample shown in Figure 12 requires the manipulator to first rotate the first joint (revolute joint about a vertical axis) in order to avoid occluding the second object. To show that our proposed methodology is able to find collision and occlusion-free paths more reliably than a standard solution, we compare it to a *bare-bones* implementation inside MoveIt!, which uses a numerical solver for the IKP (only one solution found) and no obstacle detection (virtual–occlusion pyramids and real–operator arms).

We repeat 10 times each scenario illustrated in Figure 11 and compare the performance of the two methodologies in terms of the number of attempts that result in at least a partial occlusion during the manipulator’s motion. Results are detailed in Table 7, including the success rate of each algorithm. Since our proposal considers all feasible postures for a given object’s grasping pose, we also report the number of feasible paths found, and the number which is collision-free.

In scenarios 1 to 3, the grasping manipulator starts in a vertical posture, which leads the MoveIt! numerical solver to compute an optimal path in which the arm keeps an *elbow-up* configuration, causing no collision or occlusion. Our methodology finds a similar path to the object, again without occlusion or collision. For scenarios 4 to 6, we change the initial posture of the manipulator to different folded shapes, since a cobot is unlikely to go back to an upright joint configuration after each grasp in a practical application. These postures can be seen in Figure 13. With these cases, the *bare-bones* MoveIt! version (with the numerical solution to the IKP, without virtual obstacles) is not able to reliably find an occlusion- and collision-free solution on all occasions. In fact, in scenario 5, all runs attempted with the *bare-bones* MoveIt! version fails to avoid any collision or occlusion. Meanwhile, our proposed methodology is always able to find at least one feasible path that is collision- and occlusion-free, succeeding in completing the task 8 times over 10 attempts. It should be noted that the manipulator slightly occludes one of the objects during to trials.

Finally, in scenario 6 we position the objects in such a way that it is impossible to reach the object without causing an occlusion with the camera fixed at the previous position. In this case, according to block 6 of our flow chart (Figure 2), the camera pose needs to be changed to successfully complete the task. Therefore, the camera is moved to a more favorable position, allowing our system to find occlusion-free paths. The resulting setup is shown in Figure 14. This demonstrates that our system can adapt to different scenarios by changing the pose of the camera. We have prepared the Appendix A which demonstrates how our method behaves in different scenarios in terms of repeatability and changing the camera view, compared with a bare-bones implementation of OMPL path planner.

## 10. Conclusions

In the first part of this paper, the inverse kinematic problem of a non-wrist-decoupled robot was studied using the example of the Kinova Gen3 lite manipulator. We solved it by deriving a univariate polynomial equation to find all possible values of one angle, $\theta_{1}$, then finding the corresponding values of the other joint angular positions by back substitution. The Python script used to compute the solutions to the IKP is now public. Several examples were given and compared to the solutions obtained with ROS MoveIt! IK and the real robot controller for validation. In the second part, a procedure to select the optimal solution to minimize the risk of occlusion while performing a collaborative pick-and-place task with the shortest path was proposed. The solution includes the use of a variable-pose camera to track objects within the workspace as well as the operator. Experiments to validate the procedure were included and discussed, clearly showing the usefulness of our proposal. We assessed the robustness of our algorithm based on repeatability tests in different scenarios, with varying conditions (different camera positions, different initial positions for the arm) and demonstrated that our solution was always able to find a collision-free path when possible. In a particular case where it was impossible to reach the object without causing an occlusion with the camera fixed, our method still worked by first moving the camera to a more suitable position. Compared to a standard implementation of the OMPL path planner in MoveIt!, our proposed methodology always found at least one feasible path that was collision and occlusion-free, while OMPL failed to do the same in some cases. As a Appendix A has been prepared and provided which demonstrates how our method behaves in different scenarios and compares it to a bare-bones implementation of the popular OMPL path planner library. Future work will include devising a methodology to find an optimal pose for the camera to minimize occlusion.

## Figures and Tables

**Figure 1 sensors-22-06430-f001:**
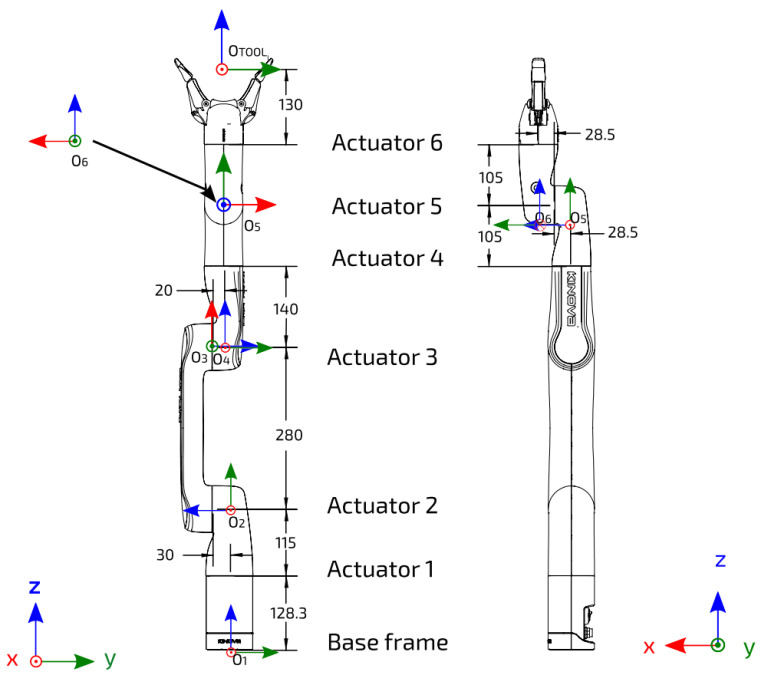
DH frames for each joint with the link dimensions (extracted from the manipulator user manual).

**Figure 2 sensors-22-06430-f002:**
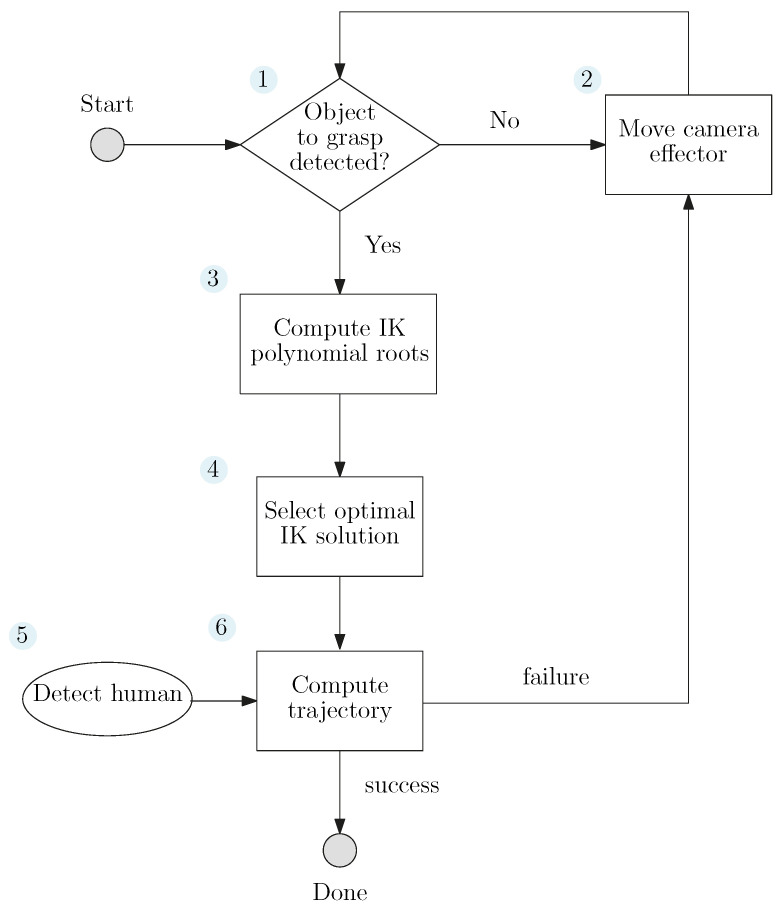
Flow chart of our occlusion-free pick-and-place solution for collaborative assembly tasks.

**Figure 3 sensors-22-06430-f003:**
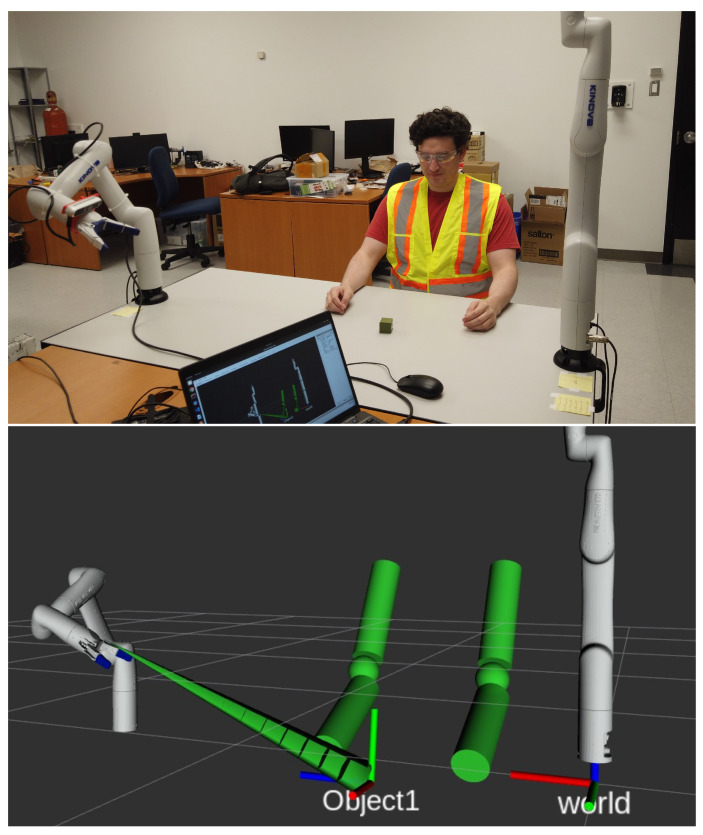
View of the experimental setup: at the **top**—a photo of the manipulators, camera, operator arms, and the target object; at the **bottom**—the visualization (in Rviz) of the same scene, with the virtual obstacles in green.

**Figure 4 sensors-22-06430-f004:**
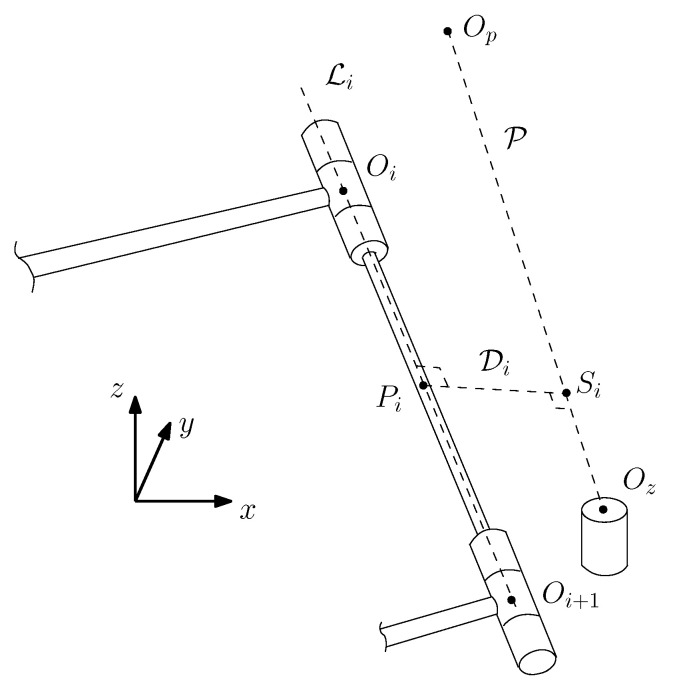
Schematic of the distance ($D_{i}$) computed between the arm link and the line of sight to the objects.

**Figure 5 sensors-22-06430-f005:**
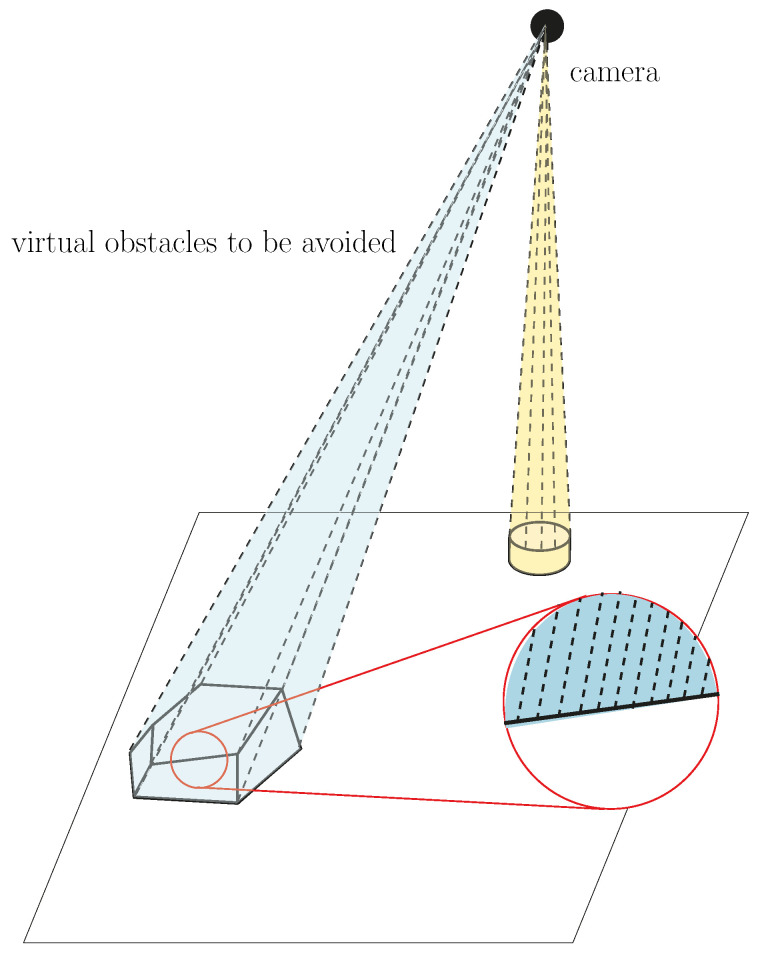
Pyramids representing the line of sight between the camera and two objects (only lines starting at the object’s vertices are shown on the left-hand side for clarity).

**Figure 6 sensors-22-06430-f006:**
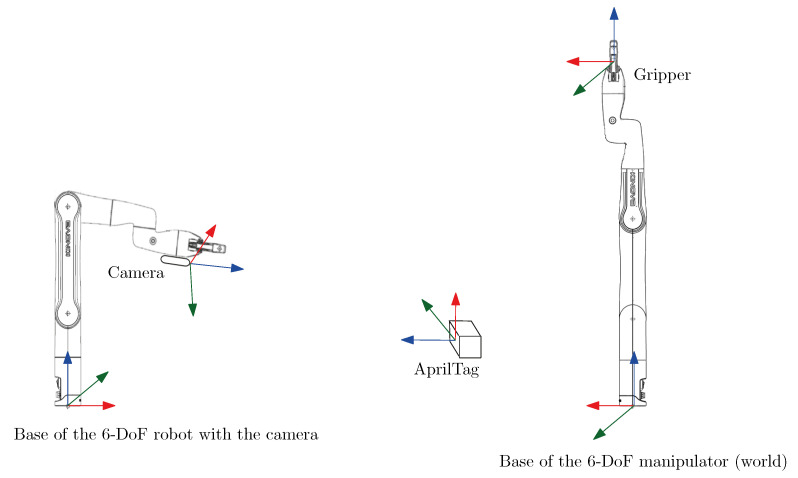
Frames in the experimental setup (*x*-axis in red, *y*-axis in green, *z*-axis in blue).

**Figure 7 sensors-22-06430-f007:**
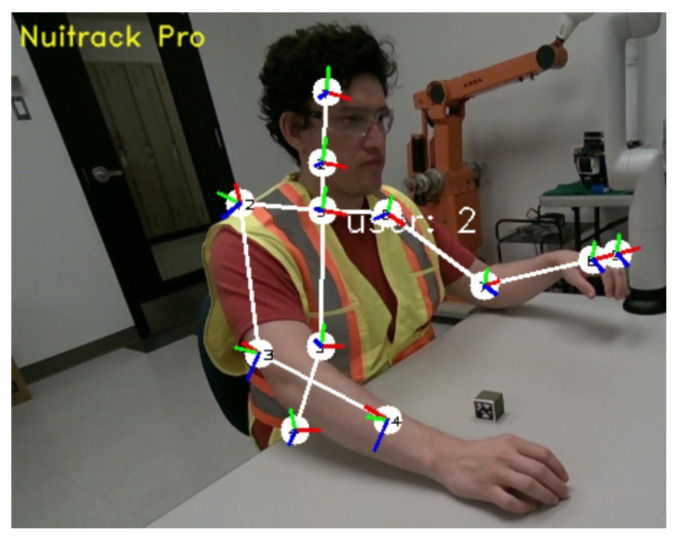
View from the Nuitrack application showing the user’s skeleton as detected by the software. Our solution extracts the joints and transfers their locations in the manipulator reference frame to generate obstacles for the path planner.

**Figure 8 sensors-22-06430-f008:**
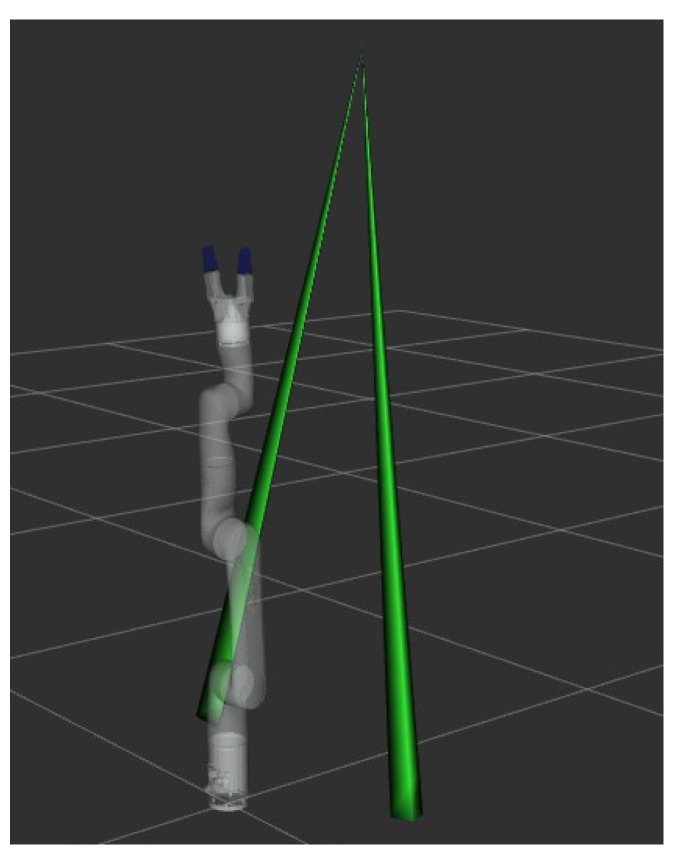
Virtual obstacles in Rviz representing the line of sight between two objects and the camera of example no. 2, Section 8.1 (the third cone between the camera and the object to be picked is not shown).

**Figure 9 sensors-22-06430-f009:**
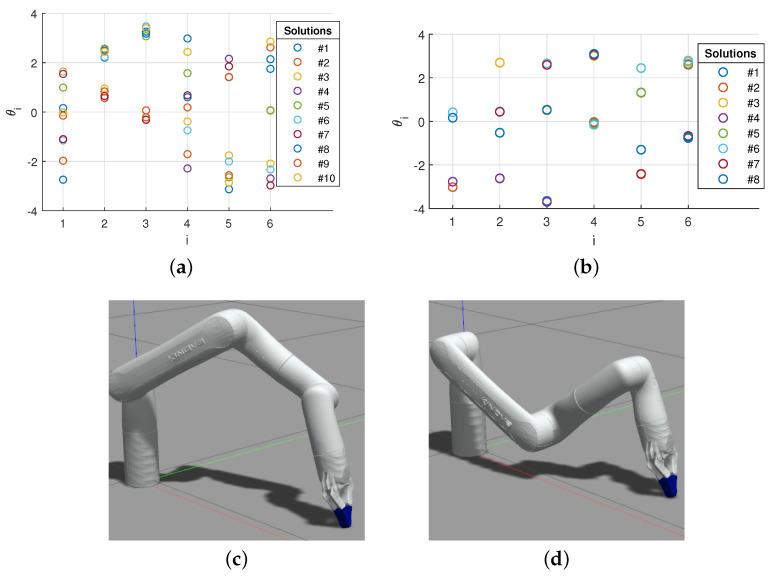
Possible postures from the examples. (**a**) Solutions to example no. 1; (**b**) solutions to example no. 2; (**c**) Ex. no. 2: solution no. 6; (**d**) Ex. no. 2: solution no. 8.

**Figure 10 sensors-22-06430-f010:**
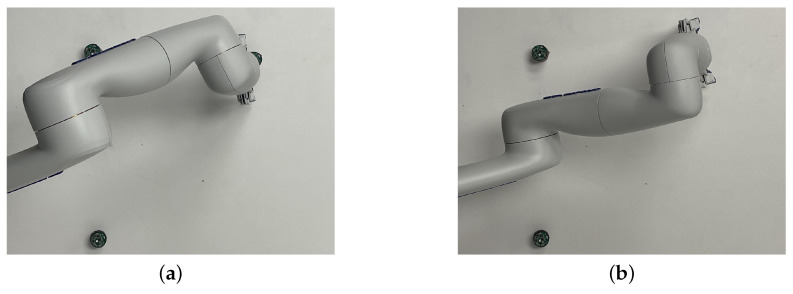
Two configurations of the arm for the same object-picking task. On the left, i.e., (**a**) solution no. 6, the other object is almost completely hidden, while the right solution, i.e., (**b**) solution no. 8, has a lot more margin.

**Figure 11 sensors-22-06430-f011:**
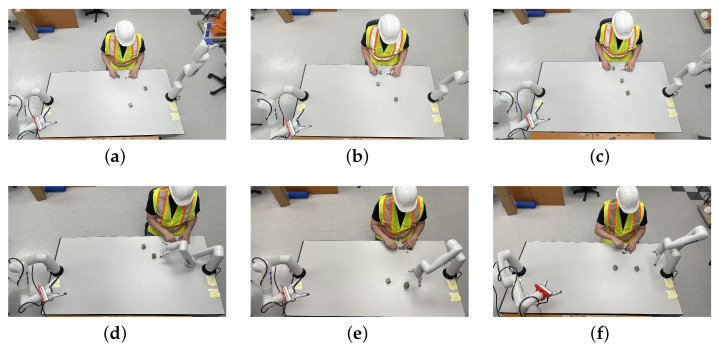
View of the positions of objects, operator, camera, and initial posture of the grasping manipulator for each of the six experimental scenarios; (**a**–**f**) show scenarios 1 to 6, respectively.

**Figure 12 sensors-22-06430-f012:**
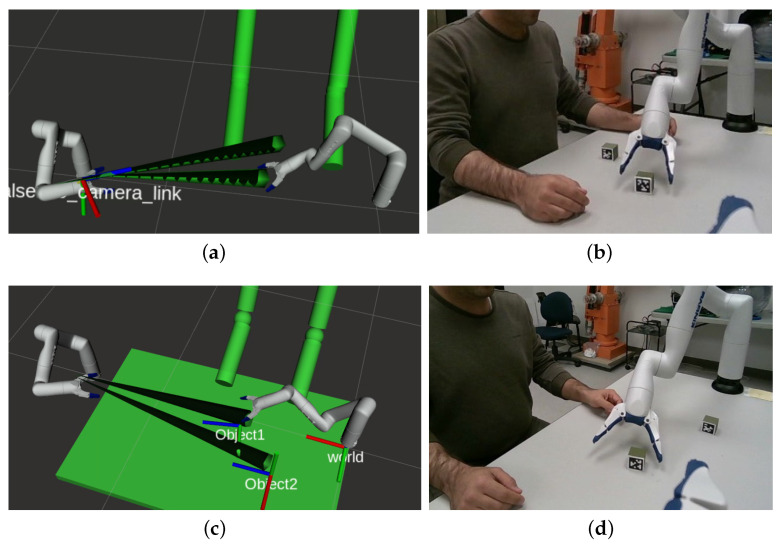
Two examples of the test cases: (**a**,**c**) are the Rviz visualizations of the virtual environment, while (**b**,**d**) are photos of the setup just before grasping the cube.

**Figure 13 sensors-22-06430-f013:**
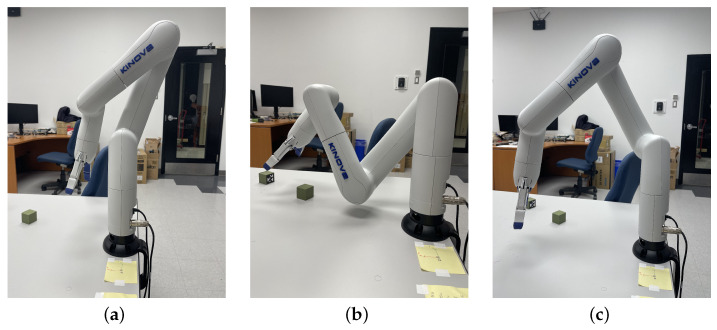
Initial positions of the grasping arm: (**a**) folded; (**b**) elbow-down; (**c**) elbow-up.

**Figure 14 sensors-22-06430-f014:**
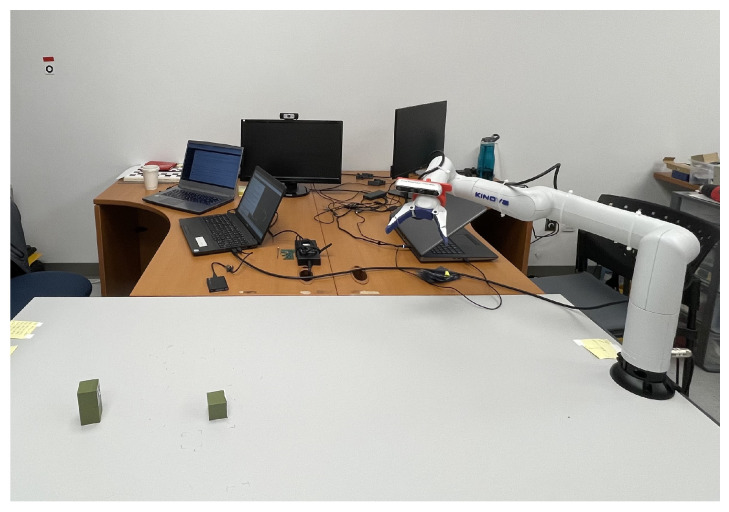
Alternative camera pose used for the sixth scenario, resulting in successful occlusion-free path-planning.

**Table 1 sensors-22-06430-t001:** DH parameters of the Kinova Gen3 lite.

i	1	2	3	4	5	6
$a_{i}$	0	$a_{2}$	0	0	0	0
$b_{i}$	$b_{1}$	$b_{2}$	$b_{3}$	$b_{4}$	$b_{5}$	$b_{6}$
$\alpha_{i}$	π/2	π	π/2	π/2	π/2	0

**Table 2 sensors-22-06430-t002:** Back substitution.

*i*	$c_{i}$	$s_{i}$
$\theta_{4}$	Equation (Equation 7)	Equation ([Disp-formula FD15a-sensors-22-06430])
$\theta_{3}-\theta_{2}$	Equation ([Disp-formula FD13a-sensors-22-06430])	Equation ([Disp-formula FD13b-sensors-22-06430])
$\theta_{5}$	Equation ([Disp-formula FD11a-sensors-22-06430]) (last row)	Equation ([Disp-formula FD11a-sensors-22-06430]) (second row)
$\theta_{2}$	Equation (Equation 4)	Equation (Equation 5)
$\theta_{6}$	Equation ([Disp-formula FD10a-sensors-22-06430])	Equation ([Disp-formula FD10b-sensors-22-06430])

**Table 3 sensors-22-06430-t003:** Numerical parameters of Kinova Gen3 lite.

*i*	1	2	3	4	5	6
$a_{i}$	0 m	0.28 m	0 m	0 m	0 m	0 m
$b_{i}$	0.243 m	0.03 m	0.02 m	0.245 m	0.057 m	0.235 m
$\alpha_{i}$	$90^{\circ }$	$180^{\circ }$	$90^{\circ }$	$90^{\circ }$	$90^{\circ }$	$0^{\circ }$
Lower limit	$-154^{\circ }$	$-150^{\circ }$	$-150^{\circ }$	$-149^{\circ }$	$-145^{\circ }$	$-149^{\circ }$
Upper limit	$+154^{\circ }$	$+150^{\circ }$	$+150^{\circ }$	$+149^{\circ }$	$+145^{\circ }$	$+149^{\circ }$

**Table 4 sensors-22-06430-t004:** Examples.

**Ex. no. 1**	***x* [m]**	***y* [m]**	***z* [m]**	ϕ **[rad]**	θ **[rad]**	ψ **[rad]**
	0.119	−0.04	0.763	−0.527	0.47	−0.759
**Ex. no. 2**	**x [m]**	**y [m]**	**z [m]**	**ϕ [rad]**	**θ [rad]**	**ψ [rad]**
	0.503	0.122	−0.002	3.077	−0.254	0.256

**Table 5 sensors-22-06430-t005:** Feasible solutions to example no. 1 (in radians).

Sol.	$\theta_{1}$	$\theta_{2}$	$\theta_{3}$	$\theta_{4}$	$\theta_{5}$	$\theta_{6}$
4	1.544	0.979	1.900	2.425	−0.982	2.021
5	0.993	1.001	1.502	0.005	0.496	−1.499
6	−1.151	0.665	1.895	−2.313	1.140	2.383
7	−1.098	−0.921	−1.885	−0.891	−1.029	1.734
8	0.160	0.910	1.609	−0.970	0.010	0.183
9	−0.145	−0.735	−1.786	−1.382	−1.718	1.049
MoveIt!	1.54	0.98	1.90	2.40	−0.98	2.00
Robot	1.59	1.00	1.93	2.39	−1.00	2.01

**Table 6 sensors-22-06430-t006:** Feasible solutions to example no. 2 (in radians).

Sol.	$\theta_{1}$	$\theta_{2}$	$\theta_{3}$	$\theta_{4}$	$\theta_{5}$	$\theta_{6}$
5	0.415	−2.010	−1.030	−1.678	−1.829	−1.444
6	0.414	−1.122	1.092	−1.733	−0.692	−1.292
7	0.166	−1.131	1.021	1.508	0.732	1.530
8	0.166	−2.091	−1.045	1.527	1.837	1.472
MoveIt!	0.40	−0.87	1.10	−1.55	−0.96	−1.05
Robot	0.45	−2.20	−1.19	−1.74	−1.76	−1.32

**Table 7 sensors-22-06430-t007:** Results of the pick-and-place experiments (each scenario is repeated 10 times).

Scenario		a	b	c	d	e	f
Our proposal	Feasible solutions	8	8	8	8	8	8
	Occlusion-free solutions	8	8	8	8	8	6
	Attempts with occlusion	0	0	0	0	2	0
	Success rate (%)	100	100	100	100	80	100
*Bare-bones* MoveIt!	Attempts with occlusion	0	0	0	2	10	6
	Success rate (%)	100	100	100	80	0	40

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
