# Peer review of "Minimize Tracking Occlusion in Collaborative Pick-and-Place Tasks: An Analytical Approach for Non-Wrist-Partitioned Manipulators"

_sensors, 2022, doi:10.3390/s22176430_

Round 1

Reviewer 1 Report

Reviewers’ Comments

Manuscript ID: sensors-1833793

Title: Minimize tracking occlusion in collaborative pick-and-place tasks: an analytical approach for non-wrist-partitioned manipulators

Authors: Hamed Montazer Zohour*, Bruno Belzile, Rafael Gomes Braga and David St-Onge

Comments:

Robotic manipulators have been widely used in industry for pick-and-place tasks. This paper presented an analytical approach for non-wrist-partitioned manipulators control in assembly lines with occlusion-free capability. The developed method has been verified by numerical simulation and experimental studies. This topic is interesting and useful for pick-and-place tasks applications in industry. However, this paper has not been well-organized. The reviewer cannot recommend the present form of this manuscript for publication in this journal.   

 Specific comments:

 (1)  The main contents of this paper are to develop theoretical method for robust control of non-wrist-partitioned manipulators in collaborative pick-and-place tasks. However, the innovative contributions have not been clearly stated in Abstract section. And this paper would be better if the proposed method is compared with the state-of-the-art control strategies in the field of robotics.

 (2)  The introduction and the literature review are too short and do not clearly highlight the contribution with to the existing work. In particular, I miss a genuine section dedicated to that, with a better analytical and critical presentation of the literature.

(3)   The writing format of this paper is more like a research report not a scientific paper, especially Section 3.4. And the structure of this paper needs to be reorganized, for example, the descriptions of the experimental setup should be combined with experiments, the analytical derivation needs to be refined, etc. In my opinion, the organization of this paper could be the introduction, theory development, numerical validation, and experimental verification.

Reviewer 2 Report

Review for the Minimize tracking occlusion in collaborative pick-and-place tasks: an analytical approach for non-wrist-partitioned manipulators submission.
The article is well written, original, addresses a current topic. It reads well, minor corrections in the English are recommended.
The article deals with tracking occlusion in a collaborative pick-and-place
task scenario. It particularly deals with the kinematics of the robots and the
methodology behind the task.
It gives a very detailed explanation of the setup and discusses several test
cases.

It is built on a particular, Kinova arm, How much would the solution change for other arms?
While the methods are sound, the robustness of the algorithm should be discussed. In this aspect, the paper could also include some references like:
- Ficuciello, F., Villani, L. and Siciliano, B., 2016. Impedance control of
redundant manipulators for safe human-robot collaboration. Acta Polytechnica, 13(1), pp.223-238.
And also Shadrin, Gennady K., et al. "Application of compensation algorithms to control the movement of a robot manipulator." Acta Polytechnica Hungarica 17.1 (2020): 191-214. Or Ogorodnikova, O., 2009. How safe the human-robot coexistence is? Theoretical presentation. Acta Polytechnica Hungarica, 6(4), pp.51-74.

The paper otherwise can be accepted after minor revision.
